# Not All Bilinguals Are the Same: A Call for More Detailed Assessments and Descriptions of Bilingual Experiences

**DOI:** 10.3390/bs9030033

**Published:** 2019-03-24

**Authors:** Angela de Bruin

**Affiliations:** Basque Center on Cognition, Brain and Language (BCBL), 20009 Donostia-San Sebastián, Spain; a.debruin@bcbl.eu

**Keywords:** bilingualism, bilingual experiences, executive functioning, language proficiency, language use, language switching, interactional contexts

## Abstract

No two bilinguals are the same. Differences in bilingual experiences can affect language-related processes but have also been proposed to modulate executive functioning. Recently, there has been an increased interest in studying individual differences between bilinguals, for example in terms of their age of acquisition, language proficiency, use, and switching. However, and despite the importance of this individual variation, studies often do not provide detailed assessments of their bilingual participants. This review first discusses several aspects of bilingualism that have been studied in relation to executive functioning. Next, I review different questionnaires and objective measurements that have been proposed to better define bilingual experiences. In order to better understand (effects of) bilingualism within and across studies, it is crucial to carefully examine and describe not only a bilingual’s proficiency and age of acquisition, but also their language use and switching as well as the different interactional contexts in which they use their languages.

## 1. Introduction

The question whether bilingualism affects executive functioning has been the focus of much recent research. For instance, bilinguals have been argued to be better at suppressing interfering information (e.g., [1]), monitoring conflict (e.g., [2]), and switching between tasks (e.g., [3]). At the same time, there are many studies that do not observe differences between bilinguals and monolinguals on various executive control tasks (e.g., [4]), with recent meta-analyses concluding that there is no systematic evidence for enhanced executive functioning in bilinguals [5]. Whether bilingualism affects executive functioning remains hotly debated. Inconsistent findings across studies and tasks may partly be related to the broadness of ‘executive functioning’, an umbrella term that encompasses different cognitive processes. In addition, task impurity is likely to play a large role. Tasks do not just measure one specific component (e.g., switching) but also have their own task-specific features that affect how participants perform.

Just like ‘executive functioning’, ‘bilingualism’ is an umbrella term too (cf. [6]). Even though bilinguals are often compared to monolinguals as two distinct groups, no two bilinguals (or monolinguals) are the same. Bilinguals can differ from each other in many different ways, including their age of acquisition, language proficiency, use, and switching practices in daily life. Two early bilinguals with a native-like proficiency in both languages can still differ tremendously in how they actually use their languages. Language-related differences between bilinguals may also be associated with their performance on executive control tasks. For instance, Prior and Gollan [7] observed that only bilinguals who frequently switch between their languages in daily life outperform monolinguals on non-verbal task-switching paradigms. In recent years, studies have therefore focused on the type of bilingual experiences that may be associated with enhanced executive functioning and it has been argued that more studies should take into account the heterogeneity of bilingualism (e.g., [8,9,10]).

In this review, I will first discuss several aspects of bilingualism that have been studied in relation to executive functioning. This overview is not meant as a systematic review or as a review of whether or not bilingual experiences affect executive functioning. Rather, it is intended to be a brief summary of the various bilingual experiences that have been studied as potential modulators. However, despite the large interest in individual differences between bilinguals, many research articles do not report the characteristics of their bilingual participant sample in sufficient detail. Providing a detailed, complete, and objective assessment of bilingual individuals is challenging. Nevertheless, if we want to better understand the effects of individual bilingual experiences (including their possible effects on executive functioning), we need to better understand who our bilingual participants are. In the second part, I will therefore discuss some of the challenges faced when describing bilingual experiences as well as some recently developed assessments. Together, this review aims to encourage researchers to use more objective and extensive assessments and to provide more detailed descriptions of their bilingual participants.

## 2. Individual Differences in Bilingualism

### 2.1. Age of Acquisition

Age of acquisition (AoA) has been the focus of many bilingualism studies, including those assessing differences in executive functioning between monolinguals, early bilinguals, and late bilinguals. Some of these studies reported that only early bilinguals, but not late bilinguals, outperformed monolinguals (e.g., [11,12]) For example, Luk and colleagues [12] classified early bilinguals as those who had started to use two languages actively before the age of 10 and found that these early bilinguals showed a smaller flanker cost (i.e., smaller inhibition cost) than monolinguals. The late bilinguals, in contrast, showed comparable flanker costs to the monolinguals. Similar results were found when age of acquisition was treated as a continuous variable. Other studies, however, showed that late bilinguals too can show benefits on executive control tasks. For instance, Pelham and Abrams [13] showed that early (AoA seven years or younger) and late bilinguals (AoA 13 years or older) performed similarly on the Attentional Control Task (ANT) and showed smaller conflict effects than monolinguals. In line with many studies not showing cognitive effects of bilingualism, however, there are also several studies reporting no differences between monolinguals and either early or late bilinguals (e.g., [14]) or no effects of age of acquisition as a continuous variable on a wide range of tasks [15].

The initial view proposed that early, highly proficient bilinguals should show the largest executive control advantages due to their prolonged experience managing two languages. In recent years, however, the opposite has also been proposed. The acquisition of a new language may be more effortful for late bilinguals than for infants who acquire two languages from birth. Later language acquisition may require more language control processes and stronger inhibition over the first language (L1; see [14]). In line with this argument, the effects of late versus early bilingualism may be task-dependent. For example, two studies have suggested that late bilingualism may mainly affect inhibitory control while early bilingualism may be more likely to affect switching [16] or conflict monitoring [17].

### 2.2. Proficiency

Age of acquisition is often confounded with proficiency such that early bilinguals also have a higher proficiency in the second language (L2), making it difficult to tear apart effects of AoA versus proficiency. However, there are several studies that have assessed proficiency effects in high and low proficiency bilinguals with a comparable language background. For example, Singh and Mishra [18] compared high and low proficiency bilinguals who had similar AoAs for both languages and had acquired their L2 (English) at school starting around the age of four (cf. also [19]). High proficiency bilinguals outperformed low proficiency bilinguals on a Stroop-like task in which participants had to look at a colour patch matching the colour of a centrally presented arrow while ignoring the patch the arrow was pointing at. These findings were interpreted as highly proficient bilinguals having enhanced goal-directed attention. However, a comparison of high versus low proficiency in groups of older adults who also did not differ in AoA showed no effects of proficiency [20]. In young adults too, several studies have not observed differences between high and low proficiency bilinguals (e.g., [14,15]).

### 2.3. Context of Language Acquisition

Differences between bilinguals also exist in terms of how they acquired their second language (e.g., in a classroom through formal instruction or through immersion) as well as with respect to the language that is used at school (cf. [21]). Although relatively less attention has been paid to the mode of language acquisition, the way a bilingual acquired their languages could affect executive functioning. Linck and colleagues [22] compared, amongst other groups, learners of Spanish who were immersed in a Spanish-speaking environment to those who learnt the language in a classroom. In one of their experiments, classroom learners outperformed the immersed learners on a Simon task, although this finding was not replicated in a second experiment comparing bilinguals living in an L1 context to those immersed in an L2 context. Still, bilinguals vary in their context of language acquisition. Early bilinguals can differ in the language(s) used at home and school (e.g., some bilinguals speak a minority language at home and a majority language at school, while others grow up in a bilingual household). Groups of late bilinguals may include bilinguals from different acquisition contexts (e.g., immersion versus classroom learning). Depending on the type of bilinguals that are studied, this may also affect comparisons between early and late bilinguals.

### 2.4. Language Use

Age of acquisition and proficiency do not necessarily reflect how and how often bilinguals use their languages. Bilinguals who acquired two languages at a young age may continue to only use one of them. Similarly, a late second-language learner may only use their L2 sporadically or may end up using the L2 as often as, or even more often than, their L1. Several studies have assessed the possible relationship between language use and executive functioning. For instance, de Bruin and colleagues [23,24] compared two groups of bilinguals to a group of monolinguals. While all bilinguals had acquired both languages during childhood and up to a very high proficiency level, only some continued to actively use both languages during later life. Across different measurements of executive control, no consistent differences were observed between the active and inactive bilingual language users [23], although language use did affect lexical processing [24]. Other studies have furthermore assessed effects of language use by using a proportion of daily non-L1 usage (e.g., [15]), the amount of a language spoken at home (e.g., [25]), or the amount of language use across different interactional contexts (e.g., [26]).

### 2.5. Language Switching and Language Context

In addition to differences in the *amount of* language use, bilinguals also differ in *how* they use their languages. Language switching is another type of bilingual experience that has been argued to affect performance on several types of executive control tasks. Focusing on non-verbal task switching, Prior and Gollan [7] found that bilinguals who frequently switched between their languages in daily life (Spanish-English bilinguals) showed smaller non-verbal task-switching costs than monolinguals. Bilinguals who did not frequently engage in daily-life language switching (Chinese-English bilinguals), however, performed the same as monolinguals. On tasks tapping into inhibitory control, such as the flanker and Simon task, frequent language switchers have also been found to outperform other bilingual groups, including a group of balanced bilinguals with low daily-life switching patterns [27]. A comparison between ‘single-language’ bilinguals and ‘dual-language’ bilinguals furthermore revealed smaller switching costs on a task-switching paradigm for dual-language bilinguals [28]. These two groups were comparable in terms of age of acquisition, language exposure/usage, and self-rated proficiency. However, while the single-language bilinguals used their languages in separate contexts, dual-language bilinguals used their two languages in the same context and reported more frequent inter- and intra-sentential switching in daily life.

These findings are in line with the recent argument that the effects of bilingualism on executive functioning may not only depend on how often bilinguals switch, but especially also on *how* they switch between languages in daily life [29,30]. In their Adaptive Control Hypothesis, Green and Abutalebi [29] describe three language contexts that come with different cognitive demands and processes. The single-language context (using the two languages separately, e.g., one language at home and one at work) is argued to demand cognitive processes such as goal maintenance and ongoing inhibition of the non-target language. The second, dual-language context (in which bilinguals use both languages in the same context, but with different speakers) is argued to require various control processes including conflict monitoring, interference suppression, selective response inhibition, and task (dis-)engagement. Language switching takes place frequently in this context. Switching also takes place frequently in the third, dense code-switching context. However, in this context bilinguals can freely switch between languages and can use an opportunistic planning approach using words that are most easily available regardless of the language. Thus, this type of switching may require relatively little cognitive control. Indeed, recent studies have suggested that freely producing words in two languages may be more efficient than having to use one language (e.g., [31]) or that free language switching may not come with a switching cost (e.g., [32]). In daily-life code switching, more nuanced distinctions can furthermore be made. For example, utterances in which the grammar and lexicon of both languages are used may require less inhibitory control than utterances following the grammar of one language with the insertion of words from the other language (cf. [33]; see [34] for a discussion of the role of conflict monitoring).

As such, when studying different groups of bilinguals on executive control tasks, the crucial comparison may not necessarily be between those who switch and between those who do not switch. Rather, the argued distinction appears to be between bilinguals who need to switch between their languages in a controlled manner in daily life versus those bilinguals who can freely switch.

## 3. Measuring Bilingual Experiences

Considering that not all bilinguals are the same and the increased interest in assessing *which* features of bilingualism may or may not be linked to enhanced executive functioning, it is becoming increasingly important to describe the type of bilingual (and monolingual) participants that are being tested. This is important for individual studies but becomes especially valuable when comparing different studies in systematic reviews or meta-analyses. For instance, several meta-analyses (e.g., [5]) not only examined overall effects of bilingualism on executive functioning but were also interested in the potential role of features such as proficiency or age of acquisition. However, as Lehtonen and colleagues [5] point out, a detailed classification of, for example, proficiency across studies is difficult because studies differ in the criteria used to classify their participants as having a high or low proficiency level. Furthermore, while most studies on bilingualism report some information about their participants’ age of acquisition and proficiency, many articles lack a more detailed description of language use and switching patterns. Surrain and Luk [35] examined how bilingual participants were described in 186 studies published between 2005 and 2015. Most articles (77%) reported the participants’ proficiency, but less than half of the articles provided objective scores. This estimation is similar to Hulstijn’s finding [36] that only 45% of 140 studies published in *Bilingualism: Language and Cognition* used objective measurements to define language proficiency. Surrain and Luk also assessed how language use was reported. The majority of studies (79%) in their overview provided some information about the languages used at home, although only 39% of studies reported this as a gradient (i.e., the proportion of time a language was used at home). Furthermore, information about the sociolinguistic context was often lacking.

Objectively and reliably measuring bilingual experiences is challenging, especially when participants speak less-frequently studied languages. Describing language use and switching in detail may be especially strenuous. However, recent years have seen several new objective measurements and questionnaires that provide more detailed descriptions of bilingual participants taking into account the role of different social and interactional contexts. Below, I discuss some of these measurements as well as the challenges faced when describing bilingual participants.

### 3.1. Age of Acquisition

Age of acquisition is almost unavoidably self-reported. However, the definition of age of acquisition (AoA) has been used in different ways. AoA can be defined as the start of language acquisition/learning (e.g., [14]) or, in the case of immigrants, as the arrival in the new country (e.g., [17]). Others categorised their bilinguals as early or late based on when they became fluent in their L2 [13] or as the age at which they started using the two languages actively on a daily basis [12]. A frequently used questionnaire (LEAP-Q [37]) asks participants to indicate both when they started acquiring their L2 as well as when they became fluent. Classifying early and late bilinguals based on the start of fluency or active language use may be a better indication of the actual onset of bilingualism. In contrast, age of acquisition defined as the start of language learning may be the onset of limited learning at school (e.g., learning to count in another language) without the language being acquired to a level needed for communicative purposes.

While there is no easy alternative, self-estimations of age of acquisition may be unreliable and could vary between participants. For example, some participants may base age of acquisition estimations on when they were first exposed to a language (e.g., by listening to music or watching television) while others may start counting from the age of formal classroom learning. Similarly, estimating the onset of fluency may largely depend on a participant’s own definition of fluency. To minimise interpretation differences between participants, it is therefore crucial to carefully explain in the questionnaire what is meant by age of acquisition. Furthermore, for some groups of participants, onset of active language use may be the easiest moment to estimate. This could especially be the case for bilinguals who started using a language when they moved to a new country or when they started using a language for educational purposes (e.g., at university).

For a comparison across studies, it is furthermore important to consider that the definition of late versus early bilingualism may vary between individual studies. Different cut-off points have been used, including before or after ten years old [12] or seven or younger versus 13 or older [13]. Furthermore, the definition of early versus late may depend on the age group that is tested. For example, in studies testing children, earlier cut-offs may be needed to compare early and late childhood bilinguals (e.g., three years old [11]). Thus, when systematically comparing findings across studies, it is important to base early versus late bilingualism on the actual age reported rather than on the labels provided by the authors of individual studies. Furthermore, to enhance comparability, I recommended to report AoA not only as the onset of learning, but also as the onset of active L2 use.

### 3.2. Proficiency

Language proficiency can refer to many different components, including production or comprehension, vocabulary, grammar, and overall fluency. Typically, proficiency is measured through self-reports asking participants to indicate on a scale (e.g., 1–7 or 1–10) how proficient they are in each of their languages. A commonly used and relatively elaborate language-background questionnaire is the LEAP-Q (Language Experience and Proficiency Questionnaire). This questionnaire, including questions about language proficiency and exposure in different settings, is currently available in 16 languages. Self-reported proficiency in this questionnaire was found to correlate reasonably well with other proficiency measurements. For example, Marian et al. [37] assessed the correlations between self-reported L1 and L2 measures in the LEAP-Q and eight measures from standardised tests (e.g., grammaticality judgements, productive vocabulary, oral comprehension). Apart from sound awareness, self-reported and standardised measurements showed moderate to high correlations in the L1 (ranging from 0.179 to 0.661) and L2 (ranging from 0.286 to 0.741). Similarly, De Bruin et al. [38] looked at correlations between self-reported proficiency and three different objective proficiency measurements (productive vocabulary, receptive vocabulary, and fluency measured in an interview) in three languages. For the two languages with larger variability in proficiency scores, correlations between self-ratings and objective measurements ranged from 0.30 to 0.66.

Language proficiency is often based on self-reported scores only, even when used to examine a more fine-grained link between proficiency and executive functioning (e.g., [14,15]). Despite their moderate to high correlations with objective measurements, self-ratings have been criticised frequently as participants may over- or under-estimate their proficiency [39]. Furthermore, self-ratings may depend on the participants’ background. Tomoschuk, Ferreira, and Gollan [40] compared self-rated proficiency to scores on a standardised picture-naming task (MINT; Multilingual Naming Test) in Chinese-English and Spanish-English bilinguals. When the results from the MINT were compared with another objective proficiency measurement (Oral Proficiency Interviews), no differences were observed between the two groups. However, the self-ratings compared to the MINT showed striking differences between the two groups of bilinguals. Focusing on Spanish/Chinese, Chinese-English bilinguals provided different self-ratings than Spanish-English bilinguals at the highest and lowest proficiency levels. That is, Chinese-English bilinguals had relatively lower MINT scores for low self-ratings but relatively higher MINT scores for high self-ratings. In English, at all self-ratings apart from the highest ones, Spanish-English bilinguals scored higher in the MINT than Chinese-English bilinguals.

The similarity between self-ratings and objective proficiency scores was also modulated by language dominance and language learning history. For example, recently immigrated Chinese speakers reported relatively lower self-rated proficiency scores while Chinese speakers who grew up in the USA but were exposed to Chinese by at least one parent provided relatively high self-ratings. This may be related to participants evaluating their proficiency against a comparison group of peers. For recent migrants, this comparison group may be Chinese speakers in China with a high proficiency level, resulting in lower self-reports. In contrast, Chinese speakers growing up in the USA may compare themselves with native English speakers with low Chinese proficiency levels, thus leading to higher self-reports.

The study by Tomoschuk et al. [40] highlights the issues that may arise when self-reported proficiency scores are compared across participants from different backgrounds, even when the same language is evaluated. Even when all studies use the same questionnaires, this is problematic for systematic reviews and meta-analyses examining effects of language proficiency across studies. However, the sole use of self-reports can also be problematic within individual studies, especially considering that bilingual participant samples often contain bilinguals speaking different languages (e.g., [14]). In addition, even when all bilingual participants speak the same languages, their background may be different (e.g., Spanish-English bilinguals living in the USA may have grown up there or may be immigrants who grew up in Mexico). Thus, only using self-rated proficiency may hinder a reliable analysis of effects of proficiency, especially when the participant sample includes bilinguals from different language backgrounds.

Picture-naming tasks can provide a more objective measurement of language proficiency that can still be administered in a short amount of time. For example, the MINT is a fast assessment using 68 pictures in increasing order of difficulty that need to be named by the participant. In this test, pictures with corresponding cognate names or pictures showing culture-specific objects were excluded. This proficiency measurement has been validated for Spanish, English, Mandarin, and Hebrew and has been found to reflect proficiency more reliably than the Boston Naming Test [41]. In terms of receptive vocabulary, the Peabody Picture Vocabulary Test (PPVT-III [42]) is another frequently used test that can provide a fast indication of English proficiency in children. Lastly, the computerised LexTALE task is another fast measurement of receptive vocabulary. This test consists of a lexical decision task that is available in multiple languages including Dutch, English, and German [43], French [44], Spanish [45], and Basque [38].

However, this is not to argue that questionnaires and self-reports should be avoided altogether. Different measurements tapping into different aspects of proficiency (e.g., production and comprehension, vocabulary, overall fluency, etc.) will best reflect the multi-dimensional nature of proficiency. For example, across four proficiency measurements (self-ratings in addition to three objective tests), de Bruin et al. [38] showed that the most optimal proficiency classification was based on all four measurements together. While there were moderate to high correlations between the individual measurements, together they provided the most complete indication of language proficiency. Thus, only using one objective proficiency measurement will only provide an indication of one specific aspect of proficiency. I therefore recommend the use of a more comprehensive battery of proficiency measurements. Depending on the research questions asked, it is advised to include more comprehensive measurements of, for example, grammar in addition to vocabulary and overall fluency tests.

The use of standardised, objective proficiency measurements may be more feasible in some bilingual populations than others. For instance, a study testing Spanish-English bilinguals will have a larger repertoire of proficiency measurements available than a study testing speakers of less-studied languages. In addition, it can be difficult to find a standardised measurement that can be used to assess both languages. Furthermore, some studies do not focus on one language combination but include speakers of many different languages, in which case it may be especially difficult to use the same standardised proficiency test for all bilinguals and for all languages. In these cases, a more extensive questionnaire tapping into different aspects of language use and proficiency (such as the ones discussed next) may be more feasible.

### 3.3. Language Use

Although many studies focus on proficiency and age of acquisition when describing their bilingual participant sample, the importance of language use is being emphasised increasingly often (e.g., [46,47]). To obtain a fast indication of daily-life language use, self-reports are commonly used. Indeed, when studies report the participants’ language use, this is often based on questions enquiring what percentage of time a participant is exposed to each language (e.g., LEAP-Q [37]) or what percentage of time a participant speaks each language (e.g., [23]). Estimating how often each language is used in daily life is difficult, but it is especially challenging considering that bilinguals do not always use their languages in the same way. Instead, language use and exposure may very much depend on the context (cf. Grosjean’s Complementarity Principle [48]). Therefore, instead of asking participants to provide an overall exposure/use score, a more reliable estimation may be achieved by asking about exposure and use in different contexts including different interlocutors (e.g., family, friends), contexts (e.g., school, media), and topics (e.g., emotions, leisure activities).

Anderson and colleagues [49] published the ‘Language and Social Background Questionnaire’ (LSBQ) that assesses language proficiency and use in different contexts. This includes questions about language use in different stages of the lifespan (e.g., primary school, high school), different contexts (e.g., home, school, religious activities), with different interlocutors (e.g., grandparents, friends), and for different activities (e.g., reading, social media, praying). For young adults, the 62 questions were found to cluster into three main factors: English proficiency, Non-English home use and proficiency, and Non-English social use (cf. [50] for results from children and older adults). The division between non-English use at home versus social use emphasises the importance of taking context into account when describing language use and proficiency. While Anderson et al. [49] focus on using the composite LSBQ score to categorise participants as bilinguals or monolinguals, their questionnaire could also be used to provide more continuous measurements of language proficiency and use in different contexts (although the current version can only be used for bilinguals and not for multilinguals).

Gullifer and colleagues [47,51] recently proposed characterising bilingual experiences in the form of high or low diversity or entropy. High language diversity refers to bilinguals who use their languages in the same social contexts in an integrated manner, which is expected to result in frequent language switching. Low language diversity refers to clearly separated language use in which one language is used in one context (e.g., home) and the other language in another context (e.g., work). This entropy score helps to compare bilinguals who mainly function in single-language contexts versus those who live in dual-language contexts. In their case, language entropy was based on questions about L1, L2, and L3 use at home, work, in social settings, and for reading and speaking. However, they also developed the ‘languageEntropy’ package [52] that allows other researchers to calculate their participants’ language diversity profile based on their own language experience questionnaires. This novel assessment provides a promising new tool to better characterise bilingual experiences in different interactional contexts that were found to not only explain language-related individual differences [47] but also differences in executive functioning or brain networks [51].

Most studies examining the reliability of self-estimated language experiences have focused on proficiency. It is possible that self-estimations of proficiency are less reliable than self-rated language use, especially when different contexts are taken into account. However, this remains an open question and further research is needed to examine how well self-ratings reflect actual daily-life language use.

### 3.4. Language Switching and Language Context

Furthermore, questionnaires have been developed to assess daily-life language switching. A commonly used questionnaire is the Bilingual Switching Questionnaire (BSWQ, [53]), including 12 questions about language switching that can be categorised into four groups: Switches to the L1, switches to the L2, contextual switches, and unintended switches. Questions about switches to the L1/L2 include statements such as: ‘When I cannot recall a word in [language A], I tend to immediately produce it in [language B]’ and ‘Without intending to, I sometimes produce the [language A] word faster when I am speaking in [language B]’. Contextual switches refer to language switches driven by a particular topic or setting and ask participants whether there are situations or topics in which they always switch languages. Lastly, unintended switches refer to language switches that participants are not aware of or do not produce consciously. This category includes questions asking participants whether it is difficult to not switch languages in a conversation. These different switching factors were found to relate differently to linguistic experiences such as proficiency as well as to cognitive measurements such as task mixing costs [54].

Following recent arguments about the importance of interactional context [29,30], researchers should describe not just the amount of language switching but also the way bilinguals switch between their languages. While previous studies have started to make the distinction between single- and dual-language contexts and have presented new methods to quantify a bilingual’s language use as more or less diverse [47], they often still overlook that language use and switching can be fundamentally different even within dual-language contexts. That is, some bilinguals may use their languages in a strict dual-language context in which they can use specific languages only with specific interlocutors (e.g., using Spanish and English at work, but always with different colleagues). Other bilinguals may instead be in a context in which they can freely use the two languages and switch when they want (i.e., when two bilinguals speak the same languages). Considering that these two types of dual-language contexts may come with completely different demands on language control (cf. [29]) it is crucial to better classify *how* bilinguals use and switch between their languages. At a minimum, studies on bilingualism should include a description of the general sociolinguistic context (cf. [35]). Questionnaires with questions about different types of daily-life switches can also help to classify whether a bilingual is more frequently switching in a free or in a more controlled manner. For example, the questions in the BSWQ about lexical-access related L1/L2 switches (i.e., switching because a word can be produced faster in the other language) may be an indication of free switching. A clearer distinction between free and controlled switching could be made by asking about the amount of intra- versus inter-sentential switching. For example, Hartanto and Zhang [28] asked their participants to indicate how often they switched languages between sentences versus how often they mixed words (within a sentence). Mixing within a sentence is unlikely to happen in a strict dual-language context in which the languages are used in the same context but with different people (cf. [29]). This rating may thus be an indicator of free code-switching. Indeed, Hartanto and Zhang found opposite correlations between task-switching costs and inter-sentential versus intra-sentential switching (i.e., the former showed a negative correlation while the latter was a positive predictor of task-switching costs). Importantly, switching estimations were requested for different contexts (home, school, work, and other). Similar to the amount of language use, bilingual switching patterns may vary dramatically depending on the context. As such, probing for estimations in specific contexts may result in more accurate self-ratings than asking for overall switching scores.

Instead of, or in addition to, asking participants to estimate how often they switch themselves, another approach could be to ask bilinguals to evaluate specific examples of switching. Hofweber and colleagues [34] used a frequency judgement task asking bilinguals to evaluate utterances including different types of switches. The participants were asked to indicate how often they encountered similar utterances in their lives. This method may provide a more nuanced evaluation of a bilingual’s daily-life switching habits without requiring them to estimate their own behaviour. Furthermore, this method allows for a more detailed evaluation of the specific types of intra-sentential switches that a bilingual encounters in daily life. Stimulus selection is very important for this type of judgement task as real examples need to be used. Furthermore, to avoid the influence of specific lexical items or grammatical structures on frequency judgements, a larger set of utterances needs to be presented. Hofweber and colleagues therefore selected utterances from existing corpora that included code switches. Thus, this way of measuring daily-life switching requires careful stimulus preparation and may also only be feasible for researchers interested in language pairs for which existing corpora with code-switched utterances are available.

Like self-reported proficiency, the reliability of self-estimations of language switching has been questioned (cf. [55]). Recent work asking bilinguals to estimate their switching frequency immediately after completing a language switching task in the lab showed that these self-ratings can be quite reliable [56]. Still, it is likely that participants are far less aware of their language switching behaviour in real life. Furthermore, considering the negative attitudes that exist towards code switching [57], some participants may underestimate their switching behaviour and self-ratings could thus also be modulated by the participant’s own attitudes towards switching. Lastly, the ability to self-rate switching behaviour may be related to a bilingual’s meta-linguistic awareness and self-monitoring and could therefore also affect potential correlations between self-rated language switching frequency and executive functioning.

Digital technologies may provide a solution to obtain more reliable assessments of language switching practices (cf. [58]). For instance, using the Ecological Momentary Assessment (EMA), bilinguals can be asked to assess their own language switching on a smartphone several times per day during a longer period of time. In particular for studies examining the more fine-grained effects of language use or switching, it would be worthwhile to collect more reliable and detailed assessments of daily-life language behaviour through EMA. Other alternatives would include using applications such as the EAR (Electronically Activated Recorder) which could be used to record brief snippets of conversations multiple times throughout the day (e.g., [59]). While this tool requires subsequent transcription of the recordings and as such is time-consuming, it has the additional benefit of providing examples of the exact type of switches made. Even though they may not be feasible to be used in all studies, at a minimum, techniques such as EMA and EAR should be used to better validate the existing self-estimations and questionnaires on language use and switching.

## 4. Conclusions

More detailed descriptions of bilingual participant samples are important for all studies on bilingualism, but especially for those that aim to examine the fine-grained effects of bilingual experiences on executive functioning. The best way to assess and describe bilingual participants may partly depend on the bilingual experiences that form the focus of a study. However, to allow for a better comparison between studies, researchers should at a minimum provide not only details about age of acquisition and proficiency, but also about how the bilinguals acquired their languages, the other languages they may speak, and the general sociolinguistic context. The way bilinguals use and switch between their languages is often neglected but forms an important part of daily language experiences. Therefore, extensive assessments should be used to better describe daily-life language use and switching in different contexts, including details about the time spent in single-language, controlled dual-language, or free code-switching contexts.

The use of (standardised) objective proficiency measurements is strongly recommended, especially when testing frequently studied languages for which these measurements are available. Furthermore, several extensive questionnaires have been developed in recent years that provide more detailed evaluations of bilingual experiences. Crucially, they are starting to take into account the different (social) contexts in which a bilingual uses their languages. I advise the use of these more fine-grained, context-specific questionnaires to better describe bilingual language use.

In the first place, these assessments should be used for a more detailed description of the bilingual participants in the Methods section. They are also important when studying the effects of bilingual experiences on executive functioning. Many studies use a categorical approach by comparing, for example, early to late bilinguals. Using bilingual language experiences as a continuous variable may better reflect the continuum of bilingualism and may do more justice to the often fine-grained differences between bilinguals. These analyses can use language entropy scores based on various measures or can include specific measures that are most closely related to the research question of interest. When multiple measures can be used in an analysis, there is a risk of cherry picking and only reporting those measures that showed a significant relationship. Therefore, researchers should decide a priori, justify, and preferably pre-register which measures of bilingual experiences to include in the analysis or should state that analyses are exploratory.

At the same time, the quest for fast but reliable assessments of different aspects of bilingualism remains ongoing. Better validations based on actual recordings of daily-life language use should be used to assess the reliability of the currently available and future questionnaires and measurements. Due to the heterogeneity of bilingual participant samples there may never be a ‘one size fits all’ approach. However, with the increased interest in individual bilingual experiences and the development of more detailed assessments, we should and can strive for better descriptions and assessments of bilingual experiences.

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
