# Peer review of "Not All Bilinguals Are the Same: A Call for More Detailed Assessments and Descriptions of Bilingual Experiences"

_behavsci, 2019, doi:10.3390/bs9030033_

Round 1

Reviewer 1 Report

This is an interesting and comprehensive overview of what is currently known about how bilingual and multilingual experience can differ.

The paper describes why such an overview is warranted given some of the recent controversies in the literature, for example, one regarding the role of bilingual experience in terms of its effects on executive control.

This review will be useful to researchers because it gathers the primary ways that individual differences in bilingual experience have been recently described and measured, and correctly emphasizes the importance of caring about these issues in bilingual research.

I only had a few thoughts/comments as I read the paper, which the author may want to address in a revision (in order of occurrence).

1) Line 37 is missing a "J"

2) Line 48, perhaps the author could introduce a paragraph break before "In this review.."

3) Line 210 - in this paragraph, there is some discussion of whether people should report AOA for first exposure to a language vs. the onset in fluency of a language across different domains.  While I understand why the author makes this distinction, as these differences do exist across questionnaires and likely in reality, I have often wondered whether people can consistently/reliably access that information about their own experiences.  That is, while it is possible to reliably reconstruct the age at which one is likely to have been first exposed to a language, can one really do that in terms of recalling the age at which they achieved fluency in XYZ domains?  My inclination is that there are just some questions that people cannot reliably or accurately answer about themselves, and that these questions should not appear on questionnaires of this sort as they may be misleading once analyzed, as different people will be answering based on different interpretations of the question, or simply inaccurate recollections.  

4) In the proficiency section, there is some discussion about whether we should value self-report vs. objective proficiency measures, where the argument is that objective is somewhat better than self-report.  However, I believe that some additional nuance may be necessary. First, not all self-report measures may be equally bad.  While I agree that self-evaluations are never perfect (e.g., I rate my own L2 production ability is outstanding vs. moderate), my inclination has always been that asking people to estimate slightly less self-evaluative things such as the percentage of time they spend using different languages, or the number of hours they spend etc might be more resistant to context/cohort effects (e.g., where people in monolingual communities might over-estimate how outstanding they are whereas people in highly bilingual communities might under-estimate how outstanding they are).  

5) Related to the above, while objective proficiency measures are certainly important, there are limitations with these too insofar as any objective proficiency test has to specifically measure something, and what that something is can differ across measures, and is by definition a specific rather than general reflection of overall bilingual experience. e.g., the MINT measures something about production whereas LexTale measures something about vocabulary knowledge.  It is possible that these two measures are correlated but that doesn't mean that each one alone is assessing something about general proficiency, or proficiency about domains not evaluated (e.g., grammatical ability).  

6) As well, it may be difficult to administer particular objective proficiency tests in a way that is comparable across languages, making it difficult to have a common metric along which to evaluate a given individual's L1 and L2 (and L3+).  Thus, unfortunately there is no silver bullet (which is likely why we still have to grapple with these issues in our field), though having a range of measures (both objective and self report) that people can uniformly use would certainly be desirable. 

To conclude, this is a nice review of the state of the art in how bilinguals differ, and I look forward to seeing it in print!

Author Response

My response is attached as a word file.

Reviewer 2 Report

I think that this work is a good review of the shortcomings that the author notices in most current works on the study of bilingualism. The exemplification and the argumentation of each point are good. 

I only noticed a typo that must be corrected: there is some content missing at the beginning of line line 37 (first page).

Author Response

I thank the reviewer for their positive evaluation and have corrected the typo.

Author Response

The response to Reviewer 3 is attached as a word file.

Reviewer 4 Report

I am extremely pleased with the quality of the paper. The paper is very well organized, contributes significantly to the field & provides extremely valuable insight into studying & obtaining data based on different methods.

Some typo errors:

Line 37 "Just", instead of "ust"

Lines 42, 214: extra space before the sentence starts

Author Response

I thank the reviewer for their positive evaluation of the manuscript. The typo has been corrected and extra spaces have been removed.